# Effect of *Paecilomyces tenuipes* Extract on Testosterone-Induced Benign Prostatic Hyperplasia in Sprague–Dawley Rats

**DOI:** 10.3390/ijerph16193764

**Published:** 2019-10-07

**Authors:** Young-Jin Choi, Eun-Kyung Kim, Meiqi Fan, Yujiao Tang, Young Joung Hwang, Si-Heung Sung

**Affiliations:** 1Division of Food Bioscience, College of Biomedical and Health Sciences, Konkuk University, Chungju 27478, Korea; choijang11@kku.ac.kr (Y.-J.C.); eunkyungkim@kku.ac.kr (E.-K.K.); fanmeiqi@kku.ac.kr (M.F.); yuanxi00@126.com (Y.T.); 2School of Bio-science and Food Engineering, Changchun University of Science and Technology, Changchun 130-600, China; 3Department of Food Science & Culinary, International University of Korea, Jinju 52833, Korea; hyj2839@naver.com

**Keywords:** *Paecilomyces tenuipes*, benign prostatic hyperplasia, testosterone

## Abstract

Benign prostatic hyperplasia (BPH) is one of the major public health concerns, which has a high prevalence rate and causes significant decline in men’s quality of life. BPH is highly related to sexual hormone metabolism and aging. In particular, dihydrotestosterone (DHT), to which testosterone is modified by 5α-reductase (5AR), has a significant effect on BPH development. DHT binds to an androgen receptor (AR) and steroid receptor coactivator 1 (SRC-1); then, it induces the proliferation of a prostate cell and expression of prostate specific antigen (PSA). *Paecilomyces tenuipes* (*P. tenuipes*) is a mushroom that has been popularized by the artificial cultivation of fruiting bodies based on silkworms by researchers from the Republic of Korea. In a previous study, we identified the effect of PE on PSA mRNA expression in LNCaP cells. This suggests that PE may have an inhibitory effect on androgen signaling. Therefore, we confirmed the expression of androgen signaling-related factors, such as AR, SRC-1, and PSA in LNCaP. Furthermore, we confirmed the androgen signaling inhibitory effect of PE using the testosterone propionate (TP)-induced BPH rat model. A BPH rat model was established with a four-week treatment of daily subcutaneous injections of testosterone propionate (TP, 3 mg/kg) dissolved in corn oil after castration. The rats in the treatment group were orally gavaged *P. tenuipes* extract (PE), finasteride (Fi), or saw palmetto extract (Saw) with TP injection. DHT induced an increase in the expression levels of AR, SRC-1, and PSA proteins in LNCaP cells. On the contrary, the PE treatment reduced the expression levels. In vivo, the BPH group showed an increase in prostate size compared with the control group. The PE gavaged group showed a decrease in prostate size compared with the BPH group. In addition, the protein expressions of AR, 5AR2, and PSA were significantly lower in the PE gavaged group than BPH group in prostate tissue. These results suggest the beneficial effects of PE on BPH via the modulation of AR signaling pathway.

## 1. Introduction

Benign prostatic hyperplasia (BPH) is the most prevalent prostate disease and is one of the representative chronic diseases caused by aging in men [1]. Half of men over 60 years of age suffer from BPH symptoms [2]. With the gradual increase in the population of the elderly, the number of BPH patients is expected to increase further. In the early stages of BPH, an enlarged prostate compresses the bladder, thereby causing lower urinary tract symptoms (LUTS), including urgency, frequency, straining, urinary intermittency, weak stream, incomplete emptying, and nocturia. As the prostate enlargement progresses, it may cause loss of bladder function and damage to renal function [3].

Normally, androgens are the most essential factors in growth and function maintenance of the prostate. In the elderly, sex hormone imbalance is considered one of the biggest causes of BPH, as androgens decrease and female sex hormones increase. However, the exact cause is not yet known. Meanwhile, according to several studies, BPH patients had higher levels of dihydrotestosterone (DHT), one of the male hormones, than normal men. Therefore, the development of BPH is closely related to androgens [1,4]. DHT is a more powerful androgen than testosterone because of its high affinity for androgen receptors (AR) [5]; 5α-reductase (5AR), an enzyme that converts testosterone to DHT, contributes significantly to the development of BPH [5]. DHT binds to an AR and steroid receptor coactivator 1 (SRC1); then, it induces the proliferation of a prostate cell [6]. DHT combines with AR and increases prostate specific antigen (PSA) expression levels in the cells; in other words, PSA levels can be increased by increasing prostate cell proliferation [7]. In addition, estrogen receptors (ER) α and β have recently been reported to be associated with the development of BPH. Estradiol, an estrogen, is converted to testosterone by aromatase in adipocytes and is involved in prostate cell proliferation and apoptosis through binding to ER [8].

The BPH patients with symptoms of LUTS require surgery or medical treatment. Currently, the most widely used drugs are 5α-reductase inhibitors (5ARIs) and alpha blockers [9]. Alpha blockers—α_1_ -adrenergic receptor antagonists—are useful in the treatment of BPH because of their ability to relax the smooth muscle of the prostate and bladder neck, which helps urine flow. However, they have a limitation in that they cannot help to relieve enlargement of the prostate [10]. The 5ARIs, such as finasteride (Fi), inhibit the conversion of testosterone to DHT because they inhibit 5α-reductase type2 (5AR2); consequently, they prevent AR binding and reduce PSA levels in BPH patients. However, 5ARIs cause side effects, including loss of appetite, ejaculation disturbances, or impotence [11,12]. Thus, alternative medicines with fewer side effects are required.

*Paecilomyces tenuipes* (*P. tenuipes*) is a mushroom that has been popularized by the artificial cultivation of fruiting bodies based on silkworms by researchers from the Republic of Korea [13]. It exhibits various physiological activities, such as anticancer, immune enhancement, and hypoglycemic activities [14]. In addition, previous studies showed that *P. tenuipes* reduced PSA mRNA expression in LNCaP cells. The studies showed that *P. tenuipes* inhibited the expression of angiogenesis-related factors and PSA in the cells [15]. PSA expression is strongly related to androgen signaling, which is correlated with the development of BPH. However, there are no studies related to BPH. Therefore, in this study, we investigated the effect of *P. tenuipes* on BPH in vivo.

## 2. Materials and Methods

### 2.1. Materials

Testosterone propionate was provided by Tokyo Chemical Industry Co. (Tokyo, Japan). Finasteride (≥ 97% pure) and dihydrotestosterone (≥ 99% pure) were purchased from Sigma–Aldrich Inc. (St. Louis, MI, USA). The DHT enzyme-linked immunosorbent assay (ELISA) kit was purchased from SunLong Biotech Co. (Hangzhou, China). Roswell Park Memorial Institute medium (RPMI), fetal bovine serum (FBS), and penicillin/streptomycin were purchased from Gibco (Big Cabin, OK, USA).

### 2.2. Sample Preparation

*P. tenuipes* was donated by the National Institute of Horticultural Herbal Science (Eumseong, Republic of Korea). The dried *P. tenuipes* were homogenized using a grinder, and 100 g of this powder was extracted by reflux with 1 L of D.W at 100 °C for 1 h. The *P. tenuipes* extract (PE) was subsequently filtered (0.25-μm pore size), freeze-dried, and then stored at −20 °C until usage. The PE extract, which was dissolved in distilled water, was served in all of the experiments.

### 2.3. Cell Culture

Human androgen-dependent prostate cancer (LNCaP) cells were purchased from the Korean Cell Line Bank (Seoul, Republic of Korea, KCLB numbers: 21740). LNCaP cells were cultured in RPMI supplemented with 100 mg/mL penicillin/streptomycin and 10% FBS. The cells were maintained in a CO_2_ incubator at 37 °C. Then, the cells were co-incubated with DHT (10 nmol) and PE (125, 250, or 500 mg/mL) for 24 h. These cells were collected for Western blotting analysis of AR, SAR2, and PSA expressions.

### 2.4. Animal Study Design

The Sprague–Dawley (SD) rats (*n* = 48, ten weeks old) were provided by Nara Biotech Co. Ltd (Pyeongtaek, Republic of Korea). The rats were placed in a specific pathogen-free (SPF) room maintained at an air-conditioned temperature (23–25 °C) and relative humidity (50–60%) on a 12h light/dark cycle. Water and feed were provided ad libitum. All animal care procedures and experiments were approved by the Institutional Animal Care and Use Committee of Konkuk University (KU19196). 

Castration was performed to rule out the effects of intrinsic testosterone, and the BPH rat model was based on the study by Coppenolle et al [16]. The rats were grouped as follows: Con, corn oil injection; BPH, TP (3 mg/kg/d)/corn oil injection; BPH + PE_L, 100 mg/kg of PE-treated BPH group; BPH + PE_H, 200 mg/kg of PE-treated BPH group; BPH + Saw, 100 mg/kg of saw palmetto extract treated BPH group; and BPH + Fi, 1 mg/kg of Fi-treated BPH group. Five groups (excluding the control group) of rats were anesthetized by intraperitoneal injection of phenobarbital (50 mg/kg) and castrated aseptically to remove both the epididymis and testes. The surgical site of the rat was periodically disinfected with povidone–iodine. Three days after surgery, castrated rats were injected subcutaneously with testosterone propionate (TP, 3 mg/kg/d) dissolved in corn oil into the nape of the rat over 28 consecutive days. The experimental groups were administered with 100 or 200 mg/kg/d of PE by oral administration for 28 days. Fi (1 mg/kg/d) and saw palmetto extract (100 mg/kg/d) dissolved in distilled water served as the positive control group. Saw palmetto is the only certified functional food for anti-prostate hypertrophy in the Republic of Korea. After fasting overnight before dissection, rats were anesthetized with ether. Blood samples were taken from the heart, and prostate tissues were immediately separated and weighed.

### 2.5. Western Blotting

Harvested LNCaP cells and homogenized prostate tissues were lysed with radioimmunoprecipitation assay (RIPA) buffer. After, insoluble material was removed by centrifugation at 13,000 RPM for 20 min at 4 °C. Next, the total concentration of the extracted proteins was determined using a BCA assay, and the lysates (30 μg protein/sample) were separated by 10% dodecyl sulfate-polyacrylamide gel electrophoresis (at 120V for 90 min) and transferred onto nitrocellulose membranes. After, blocking was performed with 10 mM Tris-buffered saline containing 5% skim milk and 0.05% Tween-20 (TBST) for 1 h at room temperature. The membranes were incubated with the various primary antibodies, namely, anti-β-actin, anti-SRC-1, anti-PSA (Santa Cruz Biotechnology, Dallas, TX, USA), anti-AR, anti-ERα (Cell Signaling Technology, Inc., Danvers, MA, USA), and anti-5AR2 (Abcam Inc., Cambridge, MA, USA) overnight at 4 °C. After the membranes were washed, they were incubated with anti-mouse IgG and anti-rabbit IgG (Cell Signaling Technology, Inc., Danvers, MA, USA) conjugated secondary antibodies (diluted to 1:5000) for 2 h at room temperature. Immunodetection was performed using an enhanced chemiluminescent horseradish peroxidase (HPR) substrate (Advansta Inc., San Jose, CA, USA). Subsequently, membranes were photographed using the Davinch–Chemi Imaging System (Davinch-K., Republic of Korea). The chemiluminescent intensities of protein signals were quantified using ImageJ 1.47v software (National Institute of Health, Sacaton, AZ, USA).

### 2.6. Serum Concentrations of DHT

Blood sera were obtained by centrifuging blood at 3000 × g for 20 min at 4 °C. The concentration of DHT in the serum was determined using an DHT enzyme-linked immunosorbent assay (ELISA) kit according to the manufacturer’s instructions. The absorbance was measured at 450 nm using a spectrophotometer.

### 2.7. Statistical Analysis

All data values are presented as mean ± standard error of the mean (SEM) derived from three independent experiments. Statistical significance was analyzed by one-way analysis of variance (ANOVA) or one-tailed Student’s t test using IBM statistical package for the social sciences (SPSS) Statistics 22 software (International Business Machines Corp., Armonk, NY, USA). Results with *p* < 0.05 and *p* < 0.01 were considered as the criteria for statistical significance.

## 3. Results

### 3.1. Effect of PE on AR, SRC-1, and PSA Expressions in the LNCaP cell 

AR, SRC-1, and PSA were key expressions in BPH. The Western blotting results indicated that DHT administration significantly increased the expression of AR, PSA, and SRC-1. However, co-treatment with DHT and PE (125, 250, or 500 μg/mL) significantly downregulated this effect (Figure 1). These results show that PE inhibits AR signaling in prostate cancer cells.

### 3.2. Effect of PE Administration on Prostate Tissue Weight in BPH Rats

The dissection procedure for the prostate tissue is shown in Figure 2A. Total prostate weight and prostate index are shown in Figure 2B,C. The prostate index is commonly used to assess the degree of development of BPH [1]. The dissection of prostate tissue showed that the size of the prostate increased compared to control group in the BPH group. The BPH group (520.72 ± 42.51) indicated a relative prostate index that was significantly higher than control group (290.03 ± 50.68, *p* < 0.01). The prostate indexes of the BPH + Saw group (490.1 ± 68.68) and the BPH + PE_L group (483.23 ± 21.56) were lower compared to the BPH group; however, there was no significant difference. The prostate indexes of the BPH + Fi group (446.28 ± 43.53) and BPH + PE_H group (463.67 ± 24.56) also indicated significant decreases compared with the BPH group (*p* < 0.05). There result showed that PE_H and Fi exhibited the most anti-proliferative effect in prostate tissue.

### 3.3. Effect of PE on the DHT Levels in BPH Rats

DHT is formed by the 5AR2 from the influence of testosterone. High levels of DHT increase gene expression, which induces growth of the prostate. Figure 3 shows the DHT concentration in serum, which indicates a significantly higher DHT level in the BPH group (59.7 ± 4.50) than in the control group (37.1 ± 2.06, *p* < 0.01). The DHT level in the BPH + PE_H (52.9 ± 2.99) and BPH + Fi (47.6 ± 2.00) groups were significantly lower than those in the BPH group (*p* < 0.05). These data suggest that PE effectively reduces the production of DHT in the prostate of BPH rats.

### 3.4. Effect of PE on Expressions of Protein in Prostate Tissue of BPH Rats

Figure 4A shows the expressions of AR, ER, 5AR2, and PSA in prostate tissue by Western blotting analysis. The protein expression of AR, ER, 5AR2, and PSA were upregulated in the BPH group compared with the control group (*p* < 0.01). However, the expressions of AR, ER, 5AR2, and PSA proteins were downregulated by PE_H and Fi when compared with the BPH group (*p* < 0.01). These findings indicate that PE inhibits the androgen signaling pathway in the prostate of BPH rats.

## 4. Discussion

BPH is one of the major public health concerns, which causes a significant decline in men’s quality of life [17]. Conventional treatments, such as surgery and pharmacological therapy, have many side effects, in addition to the long latency of BPH. Treatment for BPH can relieve the symptoms, however, there is no cure for BPH [1]. Currently, common BPH drugs use alpha blockers and 5ARIs. Alpha blockers are used to relax the smooth muscles of the prostate to rapidly improve the symptoms of obstruction [18]. However, the α-blocker does not improve the size of prostate hypertrophy. Finasteride, a 5 alpha-reductase inhibitor, is the most commonly used drug for BPH medication and reduces prostate size by 20–30% by reducing intracellular DHT levels, without decreasing testosterone levels [19]. However, finasteride medical treatments have side effects, including loss of appetite, ejaculation, or impotence [11,12]. On the other hand, natural plant products have emerged as alternative therapies because they can reduce side effects and provide therapeutic efficacy [20]. Saw palmetto (Saw) is known to be one the most effective alternative BPH medicines. Currently, in the Republic of Korea, saw palmetto is the only functional health food for treatment of prostate illnesses. Saw palmetto is safe because it has no sexual side effects compared to finasteride [21]. Mushrooms are commonly used as folk medicines in East Asia, and many research results, such as anticancer [22], anti-inflammatory. and metabolic disease-improving effects have been reported [23,24]. In a previous study, PE showed inhibitory effects of PSA expression in LNCaP cells [15]. This study was designed based on the hypothesis that the results of previous studies downregulated PSA expression by inhibiting androgen signaling in DHT-treated LNCaP cells and prostate cancer cell lines. Androgen signaling pathways also affect prostate cell growth and development. This is highly correlated with BPH. According to Kato et al. (1965), TP-treated-rats showed pathological growth of prostate tissue. As a result, TP-treated-rats have been widely used in prostate hyperplasia studies. Thus, we confirmed the effect of PE on BPH using a TP-induced BPH rat model.

Androgens play a key role in the growth and maintenance of the prostate. However, decreased secretion of testosterone due to aging increases the expression of AR in prostate cells to balance the endocrine system, allowing DHT to bind to more receptors and enlarge the prostate gland. AR and coregulators, including SRC-1, have also been associated with the development of BPH [6]. After binding to DHT and AR, binding occurs with SRC-1, and transcription occurs in the nucleus, whereby growth factor expression is upregulated and cell proliferation occurs. SRC1 is a typical type of AR adjuvant that improves development of BPH [25,26]. PSA is a representative protein whose expression is regulated by androgen signaling. AR binds to DHT, regulates gene growth and proliferation, while simultaneously upregulating PSA expression [27]. Therefore, PSA levels play an important role in diagnosing prostate disease, such as BPH, inflammation of the prostate gland, and prostate cancer. [28]. Our results suggest that PE treatments downregulate the DHT-induced increase of AR, SRC-1, and PSA protein expression in LNCaP cells. In a TP-induced BPH rat model, protein expression of AR, 5AR2, and PSA was significantly lower in the BPH + PE_L group than in the BPH group. However, the level of DHT in the prostate and the prostate weight did not decrease significantly. On the other hand, prostate index, DHT concentration, and AR pathway-related protein (AR, ER, 5AR2, and PSA) expression in prostate tissue were statistically significantly decreased in PE_H-administered rats. The effect of PE was superior to that of the Saw and was similar to Fi. From these results, we hypothesize that PE seems to suppress PSA levels by reducing the androgen signaling pathway in LNCaP cells. Further investigation is needed to prove the interaction of DHT–AR–SRC1 signaling. Regarding the expression of AR protein, it was significantly reduced below control levels in all performed experiments, and the cause of the hypersuppression of AR is in need of further investigation.

According to previous research, various mushrooms, including *Phellius linteus* and *Ganoderma lucidum*, are effective in treating BPH [29,30]. Most mushrooms contain a large amount of various physiologically active substances, such as β-glucan, flavonoids, and polyphenol. Especially, flavonoids such as apigenin and epigallocatechin-3-gallate have been reported to inhibit the activity of 5AR2 and attenuate promotion of BPH [31,32]. PE contains polysaccharide-binding protein, ergosterol, and β-sitosterol [33]. Especially, β-sitosterol, one of several phytosterols, has been show to improve BPH symptoms [34,35]. Taken together with several previous studies, the present study shows that various physiologically active substances contained in the PE inhibit the expression of 5AR2 and downregulate the androgen pathway. Consequently, PE attenuates prostate growth in BPH-induced rats by downregulating androgen signaling.

## 5. Conclusions

This study provides evidence that PE improves TP-induced BPH in rats. These effects can be attributed, at least in part, to decreased prostate levels in DHT and inhibition of AR, 5AR2, and PSA protein expression in prostate tissue. As a result, prostate size in TP-induced BPH in rats decreased. These results suggest that PE may possess potential medicinal applications and could be used as a functional food for BPH.

## Figures and Tables

**Figure 1 ijerph-16-03764-f001:**
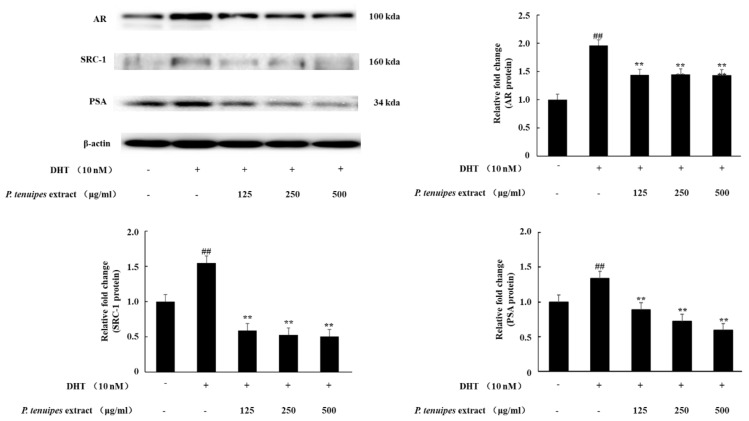
Western blotting analysis of androgen receptor (AR), steroid receptor coactivator 1 (SRC-1), and prostate specific antigen (PSA) expressions in the LNCaP cells. LNCaP cells were incubated in the medium containing dihydrotestosterone (DHT; 10 nM) and *P. tenuipes* extract (PE; 125, 250, or 500 μg/mL) for 24 h. Then, cell lysates (30 μg) were assayed for the expression levels of AR, SRC-1, and PSA by Western blotting. Data are representative of three independent experiments; data are expressed as the means ± SEM. Note: ## *p* < 0.01 compared with the DHT non-treated group; ** *p* < 0.01 compared with only the DHT treated group.

**Figure 2 ijerph-16-03764-f002:**
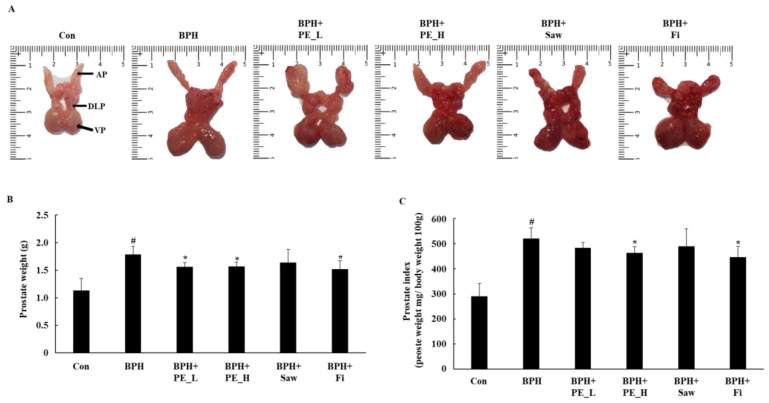
Effect of PE administration on prostate tissue weight in testosterone propionate (TP)-induced benign prostatic hyperplasia (BPH) rats. (**A**) The representatives of prostates tissue in each group; the tissue was from the dorsolateral prostate (DLP), ventral prostate (VP), and anterior prostate (AP). (**B**) The total prostate weight of the rats. (**C**) Prostate indexes. Note: Con, normal control group; BPH, TP (3 mg/kg/d)/corn oil injection; BPH + PE_L, 100 mg/kg of PE treated BPH group; BPH + PE_H, 200 mg/kg of PE treated BPH group; BPH + Saw, saw palmetto extract treated BPH group; BPH + Fi, Fi treated BPH group. Data are presented as mean ± SEM (*n* = 6). Significant differences at # *p* < 0.01 compared with the control group. Significant differences at * *p* < 0.05 compared with the BPH group.

**Figure 3 ijerph-16-03764-f003:**
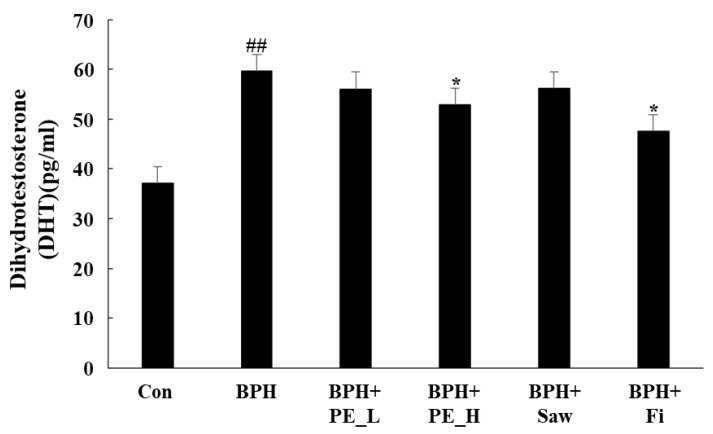
Effect of PE on the levels of DHT in serum. Individual data were obtained using a DHT enzyme-linked immunosorbent assay (ELISA). Note: Con, normal control group; BPH, TP (3 mg/kg/d)/corn oil injection; BPH + PE_L; BPH + PE_L, 100 mg/kg of PE treated BPH group; BPH + PE_H, 200 mg/kg of PE treated BPH group; BPH + Saw saw palmetto extract treated BPH group; BPH + Fi, Fi treated BPH group. Data are presented as mean ± SEM (*n* = 6). Significant differences at ## *p* < 0.01 compared with the control group. Significant differences at * *p* < 0.05 compared with the BPH group.

**Figure 4 ijerph-16-03764-f004:**
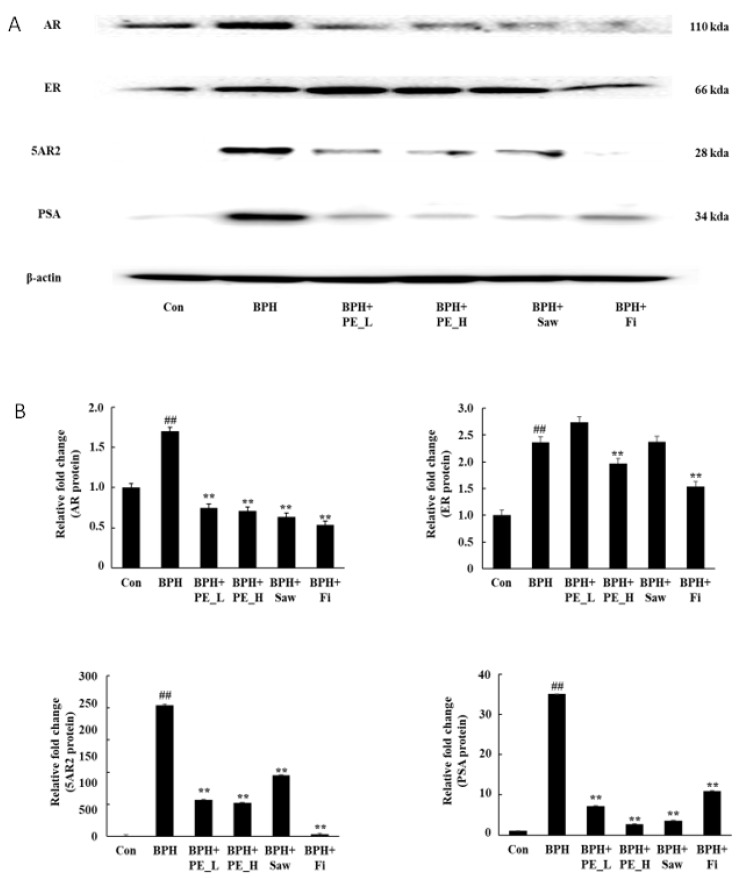
Expressions of AR, ER, 5AR2, and PSA proteins in prostate tissue. Note: Con, normal control group; BPH, TP (3 mg/kg/d)/corn oil injection; BPH + PE_L, PE low dose treated BPH group; BPH + PE_L, 100 mg/kg of PE treated BPH group; BPH + PE_H, 200 mg/kg of PE treated BPH group; BPH + Saw, saw palmetto extract treated BPH group; BPH + Fi, Fi treated BPH group. Data are presented as mean ± SEM (*n* = 6). Significant differences at ## *p* < 0.01 compared with the control group. Significant differences at ** *p* < 0.05 compared with the BPH group.

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
