# Peer review of "Effect of Paecilomyces tenuipes Extract on Testosterone-Induced Benign Prostatic Hyperplasia in Sprague–Dawley Rats"

_ijerph, 2019, doi:10.3390/ijerph16193764_

Round 1

Reviewer 1 Report

This manuscript describes the therapeutic potential of a mushroom extract, P. tenuipes(PE) on benign prostatic hyperplasia (BPH).  The subject is meaningful to the field; however, many details are missing and some of the conclusions drawn by the authors are not supported by the data shown in the manuscript.  The manuscript should not be published as it is and major revisions and further experimentation are required to address the following comments.  

Major Comments:

1) The abstract needs to be rewritten and restructured. It does not follow the normal structure of an abstract where background is given followed by the findings of the present article. Not enough background information is given upfront in the abstract. For example, the first sentence is on BPH, then the second sentence starts with LNCaP cells. This is a prostate cancer cell line, why is it used here? What is P. tenuipes (more details should be given, what is its composition, structure, where is it isolated from, etc). Why are rats treated with finasteride and saw palmetto extract? The authors need to walk the reader through the study design so that the results are clear. All abbreviations should be defined here as well as in the first mention in the text of the manuscript.

2) The statement in line 64 of the Introduction is not accurate. The authors state that they investigated the effect of the compound on BPH in vitro and in vivo. The in vitro work was conducted on a prostate cancer cell line. There are key differences between BPH and prostate cancer so the effects on BPH were not conducted. In line with this, the authors should test the effects of PE on a BPH cell line or on normal prostate cancer cells (BPH-1 or RWPE-1) and compare this the results found in LNCaP cells. This is more relevant for BPH.

3) The previous study on PE in LNCaP cells (ref 15) should be described further in the introduction. What were the findings of that published article so that the reader is clear on how are the studies presented here are different from what is already published.

4) The authors need to add more information on the rat mode so that the reader is clear on its relevance. Is it commonly used as a BPH model? Why castration is necessary in this model? Is the TP injection daily and what is the injection route (ip?, iv?). Why is saw palmetto used as a positive control? This is important since saw palmetto did not show activity in some of the current experiments.

5) The authors do not state how many times that thein vivo study was performed. Was it repeated for reproducibility? Studies in animals often show high variability so at least 2 independent trials should be performed using different extracts of PE as well.

6) The authors state that BPH has “high morbidity” (line 187). This is not accurate. BPH is a treatable disease. Not many patients die of BPH. It is true that current treatments do not cure the disease, but alleviate symptoms. This all needs to be adequately addressed.

7) The authors conclude that “PE seems to suppress PSA levels by inhibiting DHT-AR-SRC1 binding in LNCaP cells” (line 219-220). There is no evidence of this in this manuscript, therefore, this conclusion is not supported by the data shown. The authors need to test the interaction of these proteins with and without PE treatment and show the data.

8) Similarly, the authors state that “PE seems to suppress 5AR2 activity” (line 221). There no evidence of this in the manuscript either. The authors evaluated expression of 5AR2, not activity. These studies could be easily performed and included here.

Minor Comments:

1) All abbreviations should be defined upon first use. This does not include the abstract. The abstract should stand alone and abbreviations should be defined here as well.

2) The first sentence of the Introduction should site a better reference. A review article on prostatic disease should be sited here. Same for the second sentence and reference #2.

3) Line 42 of the Introduction, DHT needs to be defined here.

4) Line 47 of the Introduction says DHT incrased PSA levels. The authors need to clarify where? Do they mean PSA expression within the cells? Or PSA levels in the blood? This also needs to be clarified in line 56 of the Introduction.

5) The statement in lines 48-49 of the Introduction on estrogen receptors needs to be clarified. Why mention this here? These receptors are not studied here, so why is this relevant for this manuscript?

6) Line 51 of the Introduction, the authors list the “current” treatments for BPH, but cite a reference from 2007. This is not “current” and needs to be updated.

7) In line 67 in the Materials and Methods section, TP should be defined.

8) The Materials and Methods section should be slightly restructured. The information on where the LNCaP cells were purchased should be in the “Cell Culture” section. The details that pertain to the western blot should be in that section. This includes the information pertaining to the antibodies used (species of antibody, monoclonal versus polyclonal, etc). Also for the western blot section, the authors need to clarify that total cell lysates were made and how much of that lysate was loaded per well. General materials can be left in the Materials section.

9) Line 81 of the Materials and Methods section, information on how the compound was reconstituted after freeze-drying should be provided.

10) In Section 2.8, the authors need to clarify the following discrepancy: in this section the authors state that data are presented as the mean ±standard deviation. However, all figure legends state mean ±SEM (standard error of the mean).

11) Relevant structures of the tissues shown in Figure 2A should be added.

12) For Figures 2-4, the legends should state the actual does for the low and high doses of PE.

13) The text does not address the data shown in Figures 2A+B. The authors should add a brief sentence on this to the results section.

14) Line 159 contains a broad statement. The authors should reword this to reflect the PE has a similar effect compared to Fi in the rat model used.

15) Line 168 needs to be clarified. The authors state that DHT is one of the “major” androgens. What is meant by this? Only 10% of testosterone is converted to DHT so what is meant by “major?”

16) The authors need to clarify in line 169, that DHT levels that are shown are in the sera of treated rats.

17) The authors need to clarify line 181. What is the western blot analysis of in Figure 4? It appears to be rat prostate tissue. The methods of obtaining this tissue for western analysis is not described in the methods.

18) The first sentence of the discussion contains a lot of helpful information that should be provided as background in the introduction.

19) The authors need to clarify that PSA is an androgen-regulated gene/protein.

20) “Conjugation process of DHT” in line 209 needs to be clarified. The meaning of this phrase is not clear.

Author Response

Response to Reviewer 1Comments

This manuscript describes the therapeutic potential of a mushroom extract, P. tenuipes(PE) on benign prostatic hyperplasia (BPH).  The subject is meaningful to the field; however, many details are missing and some of the conclusions drawn by the authors are not supported by the data shown in the manuscript.  The manuscript should not be published as it is and major revisions and further experimentation are required to address the following comments.  

→ Thank you for your valuable time to review our manuscript. We are very much honored it and learned a lot from your supportive comments and useful suggestions. We also thoroughly revised whole manuscript as your comments, and our point-by-point reply to your comments was described as follows. In addition, other minor errors that we found while revising the manuscript were also corrected.

Major Comments:

1) The abstract needs to be rewritten and restructured. It does not follow the normal structure of an abstract where background is given followed by the findings of the present article. Not enough background information is given upfront in the abstract. For example, the first sentence is on BPH, then the second sentence starts with LNCaP cells. This is a prostate cancer cell line, why is it used here? What is P. tenuipes (more details should be given, what is its composition, structure, where is it isolated from, etc). Why are rats treated with finasteride and saw palmetto extract? The authors need to walk the reader through the study design so that the results are clear. All abbreviations should be defined here as well as in the first mention in the text of the manuscript.

→ Thank you very much for your valuable comments. We corrected the abstract as your comment.

2) The statement in line 64 of the Introduction is not accurate. The authors state that they investigated the effect of the compound on BPH in vitro and in vivo. The in vitrowork was conducted on a prostate cancer cell line. There are key differences between BPH and prostate cancer so the effects on BPH were not conducted. In line with this, the authors should test the effects of PE on a BPH cell line or on normal prostate cancer cells (BPH-1 or RWPE-1) and compare this the results found in LNCaP cells. This is more relevant for BPH.

→ Thank you very much for your considerable comment. We absolutely agree with you.RWPE-1 and BPH-1 cells are more suitable for experiment of BPH than LNCaP cells. However, there are a number of BHP researches evaluating the expression of androgen signaling-related proteins using LNCaP cells. The LNCaP is an androgen dependent prostate cancer cell line, and very reactive with DHT. Therefore, we would like to know whether PE could regulate androgen signaling in LNCaP cells.

Lim, S., Lee, W., Lee, D., Nam, I. J., Yun, N., Jeong, Y., ... & Kim, S. Botanical Formulation HX109 Ameliorates TP-Induced Benign Prostate Hyperplasia in Rat Model and Inhibits Androgen Receptor Signaling by Upregulating Ca2+/CaMKKβ and ATF3 in LNCaP Cells. Nutrients, 2018, 10, 1946. Kiriya, C., Yeewa, R., Khanaree, C., & Chewonarin, T. Purple rice extract inhibits testosterone‐induced rat prostatic hyperplasia and growth of human prostate cancer cell line by reduction of androgen receptor activation. Journal of food biochemistry, 2019, 43, e12987. Park, J.; Youn, D. H.; Um, J. Y. Aconiti Lateralis Radix Preparata, the Dried Root of Aconitum carmichaelii Debx., Improves Benign Prostatic Hyperplasia via Suppressing 5-Alpha Reductase and Inducing Prostate Cell Apoptosis. Evidence-Based Complementary and Alternative Medicine, 2019. Wijerathne, C. U., Park, H. S., Jeong, H. Y., Song, J. W., Moon, O. S., Seo, Y. W., ... & Kwun, H. J. Quisqualis indica improves benign prostatic hyperplasia by regulating prostate cell proliferation and apoptosis. Biological and Pharmaceutical Bulletin, 2017, b17-00468.

3) The previous study on PE in LNCaP cells (ref 15) should be described further in the introduction. What were the findings of that published article so that the reader is clear on how are the studies presented here are different from what is already published.

→ Thank you very much for your considerable comment. We described in detail about that as your comment.

4) The authors need to add more information on the rat mode so that the reader is clear on its relevance. Is it commonly used as a BPH model? Why castration is necessary in this model? Is the TP injection daily and what is the injection route (ip?, iv?). Why is saw palmetto used as a positive control? This is important since saw palmetto did not show activity in some of the current experiments.

→ Thank you very much for your considerable comment. The BPH model we used in this study was designed based ona commonly used experimental method. The castration was performed to rule out the effects of intrinsic testosterone in rats. TP were injected subcutaneously into the nape of the rat. Saw palmetto is one of the world's most famous anti-prostatic hypertrophy foods. Especially, in Republic of Korea, saw palmetto is the only anti-prostate hypertrophy functional food certified by the country. Therefore, saw palmetto was used as a positive control in this experiment.

5) The authors do not state how many times that the in vivo study was performed. Was it repeated for reproducibility? Studies in animals often show high variability so at least 2 independent trials should be performed using different extracts of PE as well.

→ Thank you very much for your considerable comment. In this study, all experiments are performed at least three independent times. We described it in the statistical analysis section.

6) The authors state that BPH has “high morbidity” (line 187). This is not accurate. BPH is a treatable disease. Not many patients die of BPH. It is true that current treatments do not cure the disease, but alleviate symptoms. This all needs to be adequately addressed.

→ Thank you very much for your valuable comment. We are sorry for our mistake. We revised the sentence as your comment.

7) The authors conclude that “PE seems to suppress PSA levels by inhibiting DHT-AR-SRC1 binding in LNCaP cells” (line 219-220). There is no evidence of this in this manuscript, therefore, this conclusion is not supported by the data shown. The authors need to test the interaction of these proteins with and without PE treatment and show the data.

→ Thank you very much for your considerable comment. We are absolutely with you. We need to do further study to prove the hypothesis. After your comment, we revised the sentence. Once again, appreciate your valuable comment.

8) Similarly, the authors state that “PE seems to suppress 5AR2 activity” (line 221). There no evidence of this in the manuscript either. The authors evaluated expression of 5AR2, not activity. These studies could be easily performed and included here.

→ Thank you very much for your considerable comment. We revised the sentence as your comment.

Minor Comments:

1)All abbreviations should be defined upon first use. This does not include the abstract. The abstract should stand alone and abbreviations should be defined here as well.

→ Thank you very much for your considerable comment. We revised it through the manuscript as your comment.

2) The first sentence of the Introduction should site a better reference. A review article on prostatic disease should be sited here. Same for the second sentence and reference #2.

→ Thank you very much for your considerable comment. We revised the references as your comment.

3) Line 42 of the Introduction, DHT needs to be defined here.

→ Thank you very much for your considerable comment. We corrected it as your comment.

4) Line 47 of the Introduction says DHT incrased PSA levels. The authors need to clarify where? Do they mean PSA expression within the cells? Or PSA levels in the blood? This also needs to be clarified in line 56 of the Introduction.

→ Thank you very much for your considerable comment. According the peripheral sentences, we described in detail to be clarified as your comment.

5) The statement in lines 48-49 of the Introduction on estrogen receptors needs to be clarified. Why mention this here? These receptors are not studied here, so why is this relevant for this manuscript?

→ Thank you very much for your considerable comment. We presented ER data in Figure 4, therefore we felled to mention it. After your mention, we described in detail ER. Hope you satisfy our description.

6) Line 51 of the Introduction, the authors list the “current” treatments for BPH, but cite a reference from 2007. This is not “current” and needs to be updated.

→ Thank you very much for your considerable comment.We replaced the reference with 2018 report as your comment.

7) In line 67 in the Materials and Methods section, TP should be defined.

→ Thank you very much for your considerable comment. We corrected it as your comment.

8) The Materials and Methods section should be slightly restructured. The information on where the LNCaP cells were purchased should be in the “Cell Culture” section. The details that pertain to the western blot should be in that section. This includes the information pertaining to the antibodies used (species of antibody, monoclonal versus polyclonal, etc). Also for the western blot section, the authors need to clarify that total cell lysates were made and how much of that lysate was loaded per well. General materials can be left in the Materials section.

→ Thank you very much for your considerable comment. We revised the Materials and Methods as your comment.

9) Line 81 of the Materials and Methods section, information on how the compound was reconstituted after freeze-drying should be provided.

→ Thank you very much for your considerable comment. We revised the sentence as your comment.

10) In Section 2.8, the authors need to clarify the following discrepancy: in this section the authors state that data are presented as the mean ±standard deviation. However, all figure legends state mean ±SEM (standard error of the mean).

→ Thank you very much for your considerable comment. We correct it as your comment.

11) Relevant structures of the tissues shown in Figure 2A should be added.

→ Thank you very much for your considerable. We described in detail about that as your comment.

12) For Figures 2-4, the legends should state the actual does for the low and high doses of PE.

→ Thank you very much for your considerable. We corrected the states as your comment.

13) The text does not address the data shown in Figures 2A+B. The authors should add a brief sentence on this to the results section.

→ Thank you very much for your considerable. Just below the Figure 2, we mentioned about that. However, after your comment, we added detailed description as your comment.

14) Line 159 contains a broad statement. The authors should reword this to reflect the PE has a similar effect compared to Fi in the rat model used.

→ Thank you very much for your considerable. We revised the sentence as your comments.

15) Line 168 needs to be clarified. The authors state that DHT is one of the “major” androgens. What is meant by this? Only 10% of testosterone is converted to DHT so what is meant by “major?”

→ Thank you very much for your considerable comment. We revised the sentence as your comment.

16) The authors need to clarify in line 169, that DHT levels that are shown are in the sera of treated rats.

→ Thank you very much for your considerable comment. We corrected it as your comment.

17) The authors need to clarify line 181. What is the western blot analysis of in Figure 4? It appears to be rat prostate tissue. The methods of obtaining this tissue for western analysis is not described in the methods.

→ Thank you very much for your considerable comment. In the western blotting section of the Methods, we mentioned that the cells and tissues were used for the assay. After your comment, we revised the sentence to be clarified.

18) The first sentence of the discussion contains a lot of helpful information that should be provided as background in the introduction.

→ Thank you very much for your considerable comment. We would like to put it again in the section because we thought it was important information. In addition, we also thought the information support the objective of our investigation. Hope you satisfy our decision.

19) The authors need to clarify that PSA is an androgen-regulated gene/protein.

→ Thank you very much for your considerable comment. PSA is a representative protein that is regulated by androgens. We revised the manuscript to clarify it as your comment.

20) “Conjugation process of DHT” in line 209 needs to be clarified. The meaning of this phrase is not clear.

→ Thank you very much for your considerable comment. We revised the sentence as your comment.

Reviewer 2 Report

This is an interesting topic with very compelling results in figure 4. 

In many areas of the manuscript, it is difficult to understand the text. I had to rely on the figures to understand the results. Therefore, I reccomend an overall re-editing of the manuscript text.

I recommend defining acronyms when they are first used in the manuscript. For example, for DH, what it stands for, dihydrotestosterone, was not defined until figure 3. It is better to define it in the introduction and abstract when they are first mentioned. 

I reccomend creating and placing the raw data (especially data from figures 1-3) into a supplementary information section in case a reader would like to take a look at the raw numbers since most of the presented results are in comparison to controls. 

On figure 1, it looks like the SRC-1 Western blots do not show up well. Did you get similar Western blot results in all of your replicate experiments for SRC-1?

In figure 4, since expression of AR protein is significantly reduced below control levels, have you all performed experiments on whether this could have negative health effects/side effects? Adding discussion on such a reduction in the discussion section of the manuscript would be interesting. Another possible discussion topic is whether you have tested PE on prostrate-derived cancer cells.

Paecilomyces are generally considered Ascomyetes and not Basidiomycetes (mushrooms). However, in the context that it was used, I am not sure if this may be just an issue with translation into English. 

Author Response

Response to Reviewer 2Comments

This is an interesting topic with very compelling results in figure 4. 

In many areas of the manuscript, it is difficult to understand the text. I had to rely on the figures to understand the results. Therefore, I reccomend an overall re-editing of the manuscript text.

→ Thank you for your valuable time to review our manuscript. We are very much honored it and learned a lot from your supportive comments and useful suggestions. We also thoroughly revised whole manuscript as your comments, and our point-by-point reply to your comments was described as follows. In addition, other minor errors that we found while revising the manuscript were also corrected. Finally, we attached the certificate of English editing.

I recommend defining acronyms when they are first used in the manuscript. For example, for DH, what it stands for, dihydrotestosterone, was not defined until figure 3. It is better to define it in the introduction and abstract when they are first mentioned. 

→ Thank you very much for your considerable comment. We corrected them through the manuscript as your comment.

I reccomend creating and placing the raw data(especially data from figures 1-3) into a supplementary information section in case a reader would like to take a look at the raw numbers since most of the presented results are in comparison to controls. 

→ Thank you very much for your considerable comment.We attached the raw data into the supplementary information as your comment.

On figure 1, it looks like the SRC-1 Western blots do not show up well. Did you get similar Western blot results in all of your replicate experiments for SRC-1?

→ Thank you very much for your considerable comment. Unfortunately, the blot was the best one. Actually, I repeated the experiment several times, but the result was the best. Hope you understand our situation.

In figure 4, since expression of AR protein is significantly reduced below control levels, have you all performed experiments on whether this could have negative health effects/side effects? Adding discussion on such a reduction in the discussion section of the manuscript would be interesting. Another possible discussion topic is whether you have tested PE on prostrate-derived cancer cells.

→ Thank you very much for your considerable comment. As you said, expression of AR protein is significantly reduced below control levels all performed experiments. We are not sure that it could have negative side effects for human health. As our knowledge, we did not find out any side effect in animal model. However, after your comment, we think that the reason of hyper-inhibition of AR is in need of further investigation, and mentioned about that in the manuscript.

Paecilomyces are generally considered Ascomyetes and not Basidiomycetes (mushrooms). However, in the context that it was used, I am not sure if this may be just an issue with translation into English. 

→ Thank you very much for your considerable comment. In Asia including Republic of Korea and China, Paecilomyces tenuipeswas known to mushroom. In addition, Ascomycota is the division or phyla of the Kingdom of mushrooms as well as Basidiomycota, form the Subkingdom of Dikarya. Therefore, Paecilomyces can be expressed in mushrooms.

Hong et al. Chemical Components of Paecilomyces tenuipes(Peck) Samson Mycobiology. 2007 Dec; 35(4): 215–218.

Round 2

Reviewer 1 Report

The authors have greatly improved the manuscript by addressing many of the reviewers’ comments. I applaud their efforts. However, due to the extensive editing that had to be completed, there are a few remaining issues that need to be further addressed prior to publication.

One major issue remains. The authors state that they investigated the effect of the compound on BPH in vitro and in vivo. The in vitro work was conducted on a prostate cancer cell line. The authors provide good reasoning for use the LNCaP cell line in that it is very responsive to DHT. I agree with their points and think this should be added to the manuscript. However, it still doesn't change the fact this is a prostate cancer cell line. Therefore, the effects on BPH were not evaluated in vitro. This statement is still not accurate. In my opinion it is still relevant to study the effects of PE on a BPH cell line or on normal prostate cancer cells (BPH-1 or RWPE-1) and compare this the results found in LNCaP cells. At the very least, the authors need to reword the sentence to more accurately describe the in vitro work.

Other minor issues remaining are as follows:

1) The abstract is significantly improved. However, now the authors have too many details in the description of the different in vivo treatment groups. It is best just to provide the general results here in the abstract and what they mean.

2) The abbreviations are much better in this version of the manuscript. However, there are still some issues. For example, Paecilomyces tenuipes is defined as “ tenuipes” in the abstract. Then the authors use “PE”, but it is not defined. This needs to be clarified so that reader is clear on what “PE” is. PSA is also not defined in the abstract.

3) The use of “con group” is really not necessary. The authors should spell out “control group” for clarity and make sure in the Materials and Methods section this control group is clearly described.

4) Adding labels to the actual figure 2A would be helpful so that the reader can see what the structure are.

5) The authors should add to the Materials/Methods section, when the prostate tissue was extracted. In other words, how long was the treatment?

6) Line 94, spell out distilled water. The abbreviation “DW” is not needed.

7) Line 221, the phrase “complete treatment” needs to be reworded. There is no cure for BPH.  

Author Response

Response to Reviewer 1 Comments

The authors have greatly improved the manuscript by addressing many of the reviewers’ comments. I applaud their efforts. However, due to the extensive editing that had to be completed, there are a few remaining issues that need to be further addressed prior to publication.

→ Thank you for your valuable time to review our manuscript. We are very much honored it and learned a lot from your supportive comments and useful suggestions. We also thoroughly revised whole manuscript as your comments, and our point-by-point reply to your comments was described as follows. In addition, other minor errors that we found while revising the manuscript were also corrected.

Major Comment:

One major issue remains. The authors state that they investigated the effect of the compound on BPH in vitro and in vivo. The in vitro work was conducted on a prostate cancer cell line. The authors provide good reasoning for use the LNCaP cell line in that it is very responsive to DHT. I agree with their points and think this should be added to the manuscript. However, it still doesn't change the fact this is a prostate cancer cell line. Therefore, the effects on BPH were not evaluated in vitro. This statement is still not accurate. In my opinion it is still relevant to study the effects of PE on a BPH cell line or on normal prostate cancer cells (BPH-1 or RWPE-1) and compare this the results found in LNCaP cells. At the very least, the authors need to reword the sentence to more accurately describe the in vitro work.

→ Thank you very much for your valuable comment. We removed ‘in vitro’ as your comment.

Other minor issues remaining are as follows:

1) The abstract is significantly improved. However, now the authors have too many details in the description of the different in vivo treatment groups. It is best just to provide the general results here in the abstract and what they mean.

→ Thank you very much for your valuable comment. We revised the abstract section as your comment.

2) The abbreviations are much better in this version of the manuscript. However, there are still some issues. For example, Paecilomyces tenuipes is defined as “ tenuipes” in the abstract. Then the authors use “PE”, but it is not defined. This needs to be clarified so that reader is clear on what “PE” is. PSA is also not defined in the abstract.

→ Thank you very much for your valuable comment. We corrected them as your comment.

3) The use of “con group” is really not necessary. The authors should spell out “control group” for clarity and make sure in the Materials and Methods section this control group is clearly described.

→ Thank you very much for your valuable comment. We revised them as your comment.

4) Adding labels to the actual figure 2A would be helpful so that the reader can see what the structure are.

→ Thank you very much for your valuable comment. We added the labes to the figure as your comment.

5) The authors should add to the Materials/Methods section, when the prostate tissue was extracted. In other words, how long was the treatment?

→ Thank you very much for your valuable comment. We had described about it in the section. If you check once again, we will very appreciate you. Thank you for your time in advance.

6) Line 94, spell out distilled water. The abbreviation “DW” is not needed.

→ Thank you very much for your valuable comment. We corrected it as your comment.

7) Line 221, the phrase “complete treatment” needs to be reworded. There is no cure for BPH.  

→ Thank you very much for your valuable comment. We revised it as your comment.
